# Identification of Antiprotozoal Compounds from *Buxus* *sempervirens* L. by PLS-Prediction

**DOI:** 10.3390/molecules26206181

**Published:** 2021-10-13

**Authors:** Lara U. Szabó, Marcel Kaiser, Pascal Mäser, Thomas J. Schmidt

**Affiliations:** 1Institute of Pharmaceutical Biology and Phytochemistry (IPBP), PharmaCampus, University of Münster, Corrensstraße 48, D-48149 Münster, Germany; lszabo@uni-muenster.de; 2Swiss Tropical and Public Health Institute (Swiss TPH), Socinstrasse 57, CH-4051 Basel, Switzerland; marcel.kaiser@unibas.ch (M.K.); pascal.maeser@swisstph.ch (P.M.); 3University of Basel, Petersplatz 1, CH-4003 Basel, Switzerland

**Keywords:** *Buxus sempervirens* L., *nor*-cycloartane alkaloids, antiprotozoal activity, multivariate data analysis, partial least squares regression, mass spectrometry, fragmentation pattern

## Abstract

Various *nor*-triterpene alkaloids of *Buxus* (*B.*) *sempervirens* L. have shown remarkable in vitro activity against the causative agents of tropical malaria and East African sleeping sickness. To identify further antiprotozoal compounds of this plant, 20 different fractions of *B*. *sempervirens* L., exhibiting a wide range of in vitro bioactivity, were analyzed by UHPLC/+ESI-QqTOF-MS/MS. The analytical profiles were investigated by partial least squares regression (PLS) for correlations between the intensity of LC/MS signals, bioactivity and cytotoxicity. The resulting models highlighted several compounds as mainly responsible for the antiprotozoal activity and thus, worthwhile for subsequent isolation. These compounds were dereplicated based on their mass spectra in comparison with isolated compounds recently reported by us and with literature data. Moreover, an estimation of the cytotoxicity of the highlighted compounds was derived from an additional PLS model in order to identify plant constituents with strong selectivity. In conclusion, high levels of antitrypanosomal and antiplasmodial activity were predicted for eight and four compounds, respectively. These include three hitherto unknown constituents of *B*. *sempervirens* L., presumably new natural products.

## 1. Introduction

Unicellular eukaryotic (“protozoan”) parasites cause enormous morbidity and mortality [1]. One of the most widespread and serious protozoan diseases is malaria.

Malaria is caused by protozoans of the genus *Plasmodium* (*P.*), which occur in erythrocytes and in liver parenchyma cells depending on their developmental stage. The disease is transmitted by insect vectors, female *Anopheles* mosquitos. Several human-pathogenic *Plasmodium* species (*P*. *falciparum*, *P*. *vivax*, *P*. *ovale*, *P*. *malariae*, *P*. *knowlesi* and *P*. *cynomolgi)* cause different clinical manifestations of the disease. The World Health Organization (WHO) estimated the annual malaria incidence at 229 million with a mortality of 409,000 cases in 2019 [2]. The human-pathogenic species *P*. *falciparum* (*Pf*) is responsible for the majority of deaths. The increasing development of resistance, especially of *Pf*, complicates the therapy and makes the development of new antimalarial drugs inevitable [3].

Protozoans of the genus *Trypanosoma* cause very severe infections considered neglected tropical diseases, namely Chagas disease in Latin America (caused by *T. cruzi*) and human African trypanosomiasis (HAT, “sleeping sickness”, caused by *T. brucei*) in sub-Saharan Africa. Although these infections are widespread and lead to severe and long-lasting illnesses, trypanosomiases still do not receive sufficient attention in drug research and development. This is mainly because they affect the poorest part of the population in already low-income countries. *Trypanosoma brucei rhodesiense* (*Tbr*) is the causative agent of East African sleeping sickness and is transmitted by the Tsetse fly (*Glossina* sp.) [4]. There are only limited therapeutic options for this medical condition, which are often associated with side effects [5,6]. Untreated, the disease has a lethality rate of 100%. Especially in view of the spread of vectors due to climate change and globalization, research into innovative and well-tolerated antiprotozoal drugs is highly relevant and future-oriented.

In our previous study, *nor*-triterpene alkaloids of the *nor*-cycloartane type from *B*. *sempervirens* L. displayed promising in vitro antiprotozoal activity and selectivity [7]. In the mentioned study, 25 alkaloids were isolated in a systematic manner from the aerial parts of *B. sempervirens* and tested for their antiprotozoal activity; several constituents with high activity against *Pf* and/or *Tbr* were isolated besides compounds with medium or low levels of activity. However, the total alkaloid fraction was found to be a highly complex mixture that might contain further alkaloids with promising activity that have not been isolated so far. Therefore, in order to identify possible further alkaloids with high antiprotozoal activity besides those already isolated, a multivariate data analysis (MVDA) approach was chosen to pinpoint the compounds responsible for the growth inhibition of the protozoa from a large amount of chromatographic/mass spectrometric data. Partial least squares regression (PLS) was used to correlate the complex UHPLC/+ESI-QqTOF-MS/MS (henceforth termed “LC/MS”) data of 20 different alkaloid fractions from a *B*. *sempervirens* L. leaf extract with their antiprotozoal bioactivity data. PLS regression has already been used in previous work by our group as well as others to make accurate predictions of constituents with antitrypanosomal [8,9,10] or antiplasmodial [11] activity, respectively, in plant extracts and fractions thereof. In this work, we present PLS models that highlight, besides those already isolated, further alkaloids of *B. sempervirens* L. with high predicted activity against *Pf* and *Tbr*. Furthermore, a corresponding model was generated for cytotoxicity against mammalian cells, to obtain some information on selectivity.

## 2. Results and Discussion

### 2.1. Antiprotozoal Activity of Crude Extract and Fractions from B. sempervirens L.

The alkaloids (ALOF) of *B*. *sempervirens* L. leaves were enriched from crude dichloromethane extract (GBUS) by acid-base extraction and subsequently separated by CPC into 20 different fractions [12]. The samples were tested in vitro against *Pf* and *Tbr* as well as against mammalian control cells (L6 rat skeletal myoblasts) to evaluate the cytotoxicity (Cytotox.) of the fractions (Table 1).

The dichloromethane extract (GBUS) and the two subfractions LNB (lipophilic and neutral constituents) and ALOF (alkaloids) showed clear differences in their antiprotozoal activity in the in vitro assays (Table 1). The alkaloid fraction very clearly displayed the highest activity against both pathogens, *Pf* and *Tbr,* as well as considerable antiplasmodial selectivity with a selectivity index (SI) of 34. The antiplasmodial activity of the alkaloid fraction was most pronounced with an IC_50_ value of 0.99 µg/mL. The growth inhibition against *Tbr* with an IC_50_ value of 5 µg/mL was weaker but still interesting. The strong increase of activity and selectivity of the alkaloid rich fraction (ALOF) in comparison with the crude extract and, particularly, with the lipophilic and neutral fraction (LNB) implied that the alkaloids are mainly responsible for the antiprotozoal activity of the dichloromethane extract of *B*. *sempervirens* L. Therefore, the alkaloid fraction was fractionated by CPC for further study.

The 20 CPC fractions displayed a wide range of activities (Table 1) in the antiprotozoal in vitro tests while the cytotoxicity against L6 rat-skeletal myoblasts, also tested as a marker for selectivity, was generally rather low and did not vary to a significant degree, with IC_50_ values ranging from 12 µg/mL in case of fraction 4 to 55 µg/mL in case of fraction 20.

Regarding activity against *Tbr*, fraction 2 was the most active with an IC_50_ value of 0.63 µg/mL. Fraction 3 and fraction 12 also exhibited IC_50_ values below 1 µg/mL. Fraction 18 was the least active against *Tbr* with an IC_50_ value of 39.3 µg/mL.

Several of the fractions (fractions 2, 3, 4, 5, 7–13) displayed a high level of antiplasmodial activity against *Pf* with IC_50_ values below 1 µg/mL. Fraction 2 was the most potent fraction also in this case, with an IC_50_ value of 0.35 µg/mL. In addition, this fraction had a high SI of 43 against *Pf*. For the other mentioned active fractions, the concentration range of antiplasmodial IC_50_ values was also distinctly below that of cytotoxicity (all SI > 10). This suggests a selective inhibitory effect of the *Buxus*-alkaloids, as opposed to an unspecific cytotoxic effect.

### 2.2. PLS Modeling

Correlations between analytical fingerprint data of complex mixtures, such as extracts and fractions and their bioactivity data can be investigated by PLS regression and the resulting models used to identify constituents mainly responsible for the biological effect under study [8,9,10]. In order to apply such an approach to the present set of fractions, they were all analyzed by LC/MS (Appendix A) and the fingerprint data were processed using the software DataAnalysis 4.1 and ProfileAnalysis 2.1 (Bruker Daltonik GmbH, Bremen, Germany). The resulting data matrix (“bucket table”) of variables representing MS signal intensity within defined retention time and *m*/*z* value intervals consisted of 1420 [*m*/*z*: t_R_] variables × 20 analyses. These data constituted the X-matrix (independent variables) while the respective negative logarithmic IC_50_ values (pIC_50_) of the fractions were treated as the Y-matrix (dependent variables) to generate PLS regression models in SIMCA 16.0.1 (Sartorius Stedim Data Analytics AB, Umeå, Sweden). Three different PLS models were calculated, namely, for the activity against *Pf* and *Tbr* as well as for cytotoxic activity against L6 rat skeletal myoblasts used as mammalian control cells to assess selectivity. High values for R^2^ (coefficient of determination of values predicted during model calibration) and, particularly, for Q^2^ (coefficient of determination of values predicted in leave-one-out cross validation) obtained with few PLS components in each case indicated that stable models with good internal predictive power had been generated (Table 2).

#### 2.2.1. PLS-Prediction of Antiplasmodial Compounds from *B. sempervirens* L.

The PLS model for antiplasmodial activity consisted of two PLS components and yielded R^2^ and Q^2^ values of 0.9 and 0.82, respectively (Table 2). Scores and loadings plots are presented in Figure 1. In the scores plot of the PLS, the different fractions from *B. sempervirens* L. are grouped along their PLS components, which form the axes of the coordinate system. The influence of individual X variables (“buckets”) on the samples’ positions in the scores plot is reflected in the loadings plot. For instance, if a fraction is located in the upper right quadrant of the scores plot (high values on both, the first and the second PLS component), the buckets responsible for the position of this fraction are found in this area of the loadings plot. At the same time, these variables are closest to the position of the Y variable, which means that they are most strongly correlated with activity.

#### 2.2.2. Identification of the Antiplasmodial Compounds Highlighted by the PLS Model

In the loadings plot, four bucket variables are emphasized as mainly important for the antiplasmodial activity of the alkaloid fractions (Figure 2). In order to assign and identify these compounds, the chromatographic and mass spectral data of the LC/MS measurements were evaluated (Table 3). The +ESI-QqTOF MS/MS spectra corresponding to the highlighted variables were examined in particular for the occurrence of prominent core fragments, which initially provide information about the substitution present and the occurrence of double bonds in the structures of the compounds. In previous studies, characteristic core fragments of Buxaceae alkaloids have already been identified for the amino-/amidosteroids in *Pachysandra* [8] and *Sarcococca* [13]. For the *nor*-triterpene alkaloids of the *nor*-cycloartane type, occurring in the genus *Buxus*, a few prominent fragments were also discovered, which can be used for structural assignment [14]. These characteristic diagnostic fragments were supplemented in the present work by further ones as shown in Figure 3.

Based on the MS chromatograms and the fragmentation pattern, as well as by identity of the retention times with samples of the authentic, isolated alkaloids [7], compounds **1** (4.98 min: 497.416 *m/z*) (Appendix A) and **4** (3.09 min: 445.388 *m/z*) (Appendix A) were identified as *O*-tigloylcyclovirobuxeine-B (**1**) and cyclomicrophylline-A (**4**), respectively, both isolated in our previous study and proven to be highly active against *Pf* in the in vitro assay (IC_50_: 1.05 µM (**1**); IC_50_: 1.76 µM (**4**)) [7]. The finding of these two compounds with already proven antiplasmodial activity can be taken as evidence for the validity of the calculated PLS model.

The occurrence of a core fragment at *m/z* 323 allowed assignment of compound **2** (4.33 min: 501.412 *m/z*) as a C-3/C-20 diamine with an additional single substitution or unsaturation in the core skeleton (Figure 3). The fragment *m/z* 401 indicated the neutral loss of a C_5_-acid moiety (100 Da; a tiglic acid ester is assumed in analogy to compound **1** from the [M + H]^+^ ion of **2**. The presence of two monomethylated amino groups at C-3 and C-20 was evident from the dual neutral losses of 31 Da (*m/z* 385 → 354 → 323) (Appendix A). The detailed fragmentation pathway is reported in Appendix A. The structure, thus determined from the mass spectral data (Figure 2), could not be found in the literature, so that we assume, to the best of our knowledge, that compound **2** is a new natural product. It should be noted that the stereochemistry of **2**, which is not accessible on grounds of the mass spectra alone, is assumed in analogy with related compounds of known configuration (e.g., reported in [7]). The generic name of the new compound, *O*-tigloylcyclomicrophylline-d, was chosen based on the existing classification for *Buxus*-alkaloids [15,16].

Compound **3** (14.73 min: 310.312 *m/z*) displayed a completely different chromatographic and mass spectral behavior from the *Buxus*-alkaloids (+ESI-QqTOF MS/MS spectra see Appendix A). Furthermore, the molecular formula C_20_H_39_NO with only two double bond equivalents is not compatible with a triterpenoid alkaloid and does not match any other known constituent from *B*. *sempervirens* L. According to the data, it could be an oleamide structure, but this was not investigated further in this study. The identification and clarification of this compound’s contribution to the antiplasmodial activity must await its isolation in subsequent studies.

#### 2.2.3. PLS-Prediction of Antitrypanosomal Compounds from *B. sempervirens* L.

The PLS model for antitrypanosomal activity is composed of four PLS components and shows good statistical performance with R^2^ and Q^2^ values of 0.98 and 0.89, respectively (Table 2). The resulting scores and loadings plots are depicted in Figure 4.

#### 2.2.4. Identification of the Antitrypanosomal Compounds Highlighted by the PLS Model

In the loadings plot of the PLS model, eight buckets were located in the upper right quadrant and thus predicted to be mainly responsible for the antitrypanosomal activity (Figure 5). The data of the LC/MS analysis of the underlying compounds are listed in Table 4.

Cyclomicrophylline-A (**4**) (3.09 min: 445.388 *m/z*) was also found as an active compound in the model against *Pf* (see Section 2.2.2 above, Figure 2, Table 3) and had already demonstrated activity in the in vitro assay against *Tbr* with an IC_50_ value of 2.3 µM [7]. In addition, compound **11** (3.55 min: 415.375 *m/z*) could be assigned (Appendix A) to cyclovirobuxeine-B, which had also been isolated and displayed promising antitrypanosomal activity (IC_50_: 1.5 µM) in our previous study [7]. These findings, as in the case of the antiplasmodial model above, corroborate the validity of the calculated model.

Compound **8** (2.30 min: 443.371 *m/z*) exhibited analytical data analogous to compound **11** (Appendix A). The structural formula C_28_H_46_N_2_O_2_ derived from the [M + H]^+^ ion (*m/z* 443) differed in the occurrence of additional oxygen and carbon atoms in comparison to **11**. According to these mass spectral data compound **8** could be assigned to the known *Buxus*-alkaloid *N*-formylcyclovirobuxeine-B. This alkaloid has already been isolated from *Buxus malayana* Ridl. [17]. Antitrypanosomal activity of this compound has not been reported so far.

Compound **6** (4.67 min: 483.402 *m/z*) showed a characteristic fragmentation pattern (Appendix A) almost identical to compound **1**. In contrast to *O*-tigloylcyclovirobuxeine-B (**1**), the presence of a primary amino group could be concluded in the structure of **6** from the neutral loss of NH_3_ observed in the fragment at *m/z* 466 [483-NH_3_]^+^. The detailed fragmentation pathway is reported in Appendix A. Based on the fragmentation, it could not be determined whether the primary amino group is present at position 3 or 20. To the best of our knowledge, neither structure has been described in the literature so that **6** certainly represents a new natural product. In accordance with the systematic naming of such alkaloids [15,16], this compound must be either *O*-tigloylcyclovirobuxeine-E or F (E = R_1_: N(CH_3_)_2_; R_2_: NH_2_; F = R_1_: NH_2_; R_2_: N(CH_3_)_2_).

The core fragments *m/z* 323 and 297 in the +ESI-QqTOF MS/MS spectrum of compound **10** (4.74 min: 385.364 *m/z*) (Appendix A) indicated a C-3/C-20 diamine with a double bond between C-6 and C-7 (Figure 3). The fragments *m*/*z* 354 [M + H-CH_3_NH_2_]^+^ and 323 [354-CH_3_NH_2_]^+^ indicated a substitution with two monomethylamino groups. The resulting structure (Figure 5) is, to the best of our knowledge, so far unknown.

Compounds **5** (3.51 min: 370.317 *m/z*) (Appendix A), **7** (2.51 min: 302.251 *m/z*) (Appendix A) and **9** (4.71 min: 354.321 *m/z*) (Appendix A) did not display the characteristic mass spectral behavior of typical *Buxus*-alkaloids so that they could not be structurally elucidated based on their fragmentation. Their identification, hence, as in the case of **3**, must await further studies.

#### 2.2.5. PLS-Prediction of Cytotoxic Compounds from *B. sempervirens* L.

Although the cytotoxic activity of the alkaloid fractions with good activity against the protozoan parasites was usually rather low and their selectivity indices high, especially in the case of antiplasmodial activity, it was nevertheless interesting to create a PLS model for the cytotoxicity data of the alkaloid fractions. In this case, a model consisting of three PLS components resulted, which yielded good statistical quality with an R^2^ of 0.96 and a Q^2^ of 0.83 (Table 2). The scores and loadings plots (Figure 6) showed the distribution of fractions and buckets according to their effect on cytotoxicity, respectively.

Of the 11 compounds with predicted antiprotozoal activity, the cytotoxicity model included compounds **7** (2.51 min: 302.251 *m/z*), **9** (4.71 min: 354.321 *m/z*), **10** (4.74 min: 385.364 *m/z*) and **11** (3.55 min: 415.375 *m/z*), which were identified as the four compounds with the greatest impact on cytotoxicity (Figure 7).

This might suggest that the antitrypanosomal activity of these four compounds could be due to an unspecific cytotoxic effect rather than a selective impact on the parasites. However, the cytotoxicity of compound **11** (3.55 min: 415.375 *m/z*) against the same L6 cell line was already tested in our previous study [7] and it was much lower than the antitrypanosomal activity (IC_50_ value against *Tbr* of 1.5 µM vs. 35.5 µM for cytotoxicity against L6 cells; SI: 24). With some caution, this result may probably be extrapolated to the other three compounds dominating the PLS model for cytotoxicity. Thus, it can be concluded that even if these four ingredients have a significant influence on the cytotoxicity of the boxwood leaves, they may still possess a selective antiprotozoal effect since the overall cytotoxicity is rather low.

## 3. Materials and Methods

### 3.1. Plant Material

The collection, identification and other details of the plant material used in this study were reported previously [12].

### 3.2. Extraction and Fractionation of B. sempervirens L. Leaves

The extraction of plant material, the acid-base extraction to obtain the alkaloid fraction (ALOF) and the separation of ALOF into 20 different fractions by centrifugal partition chromatography (CPC) was described previously [12].

### 3.3. UHPLC/+ESI-QqTOF-Mass Spectrometry

LC/MS measurements for multivariate data analysis: The previously described method and parameters were used to obtain these data [12]. The 20 fractions were analyzed in a continuous sequence to obtain optimal comparability of retention and intensity data as needed for multivariate data analysis. The sample concentration of all fractions was 2 mg/mL.

LC/MS measurements to identify constituents of particular interest: To investigate in more detail the characteristic MS fragmentation of several constituents with predicted antiprotozoal activity (i.e., after analysis of the PLS models), a separate experiment was performed in MRM mode, for which the respective [M + H]^+^ ion (deviation *m/z* ± 10) was selected beforehand. The collision energy was 40 eV. Based on these spectra, fragmentation pathways were reconstructed (Appendix A). The mass deviation of the fragments taken into consideration was <40 ppm in all cases.

### 3.4. Pretreatment of LC/MS Data

The raw LC/MS data were converted into molecular features using the find molecular features function of the software Data Analysis 4.1 (Bruker Daltonik GmbH, Bremen, Germany). The parameters were set as follows: signal/noise threshold: 2, correlation coefficient threshold: 0.7, minimum compound length: 10 spectra, smoothing width: 1, additional smoothing: yes, chemistry: positive adducts M + H, M + NH_4_, M + Na, M + K, M-H_2_O + H, 2M + H, 2M + Na, 2M + NH_4_, 2M + K, 2M + CH_3_CN + H, 2M + CH_3_CN + Na, spectrum type: line spectra only, background subtraction: none and add fragment spectra: no.

For further processing, the obtained molecular features were imported into the software ProfileAnalysis 2.1 (Bruker Daltonik GmbH, Bremen, Germany). A descriptor table (“bucket table”) was created in which each bucket value corresponds to mass signal intensity at a given *m/z* value and a given retention time [min]. The following parameters were applied: data selection and processing: find molecular features, retention time range: 1–15 min, mass range: 100–1500 *m/z*, advanced bucketing: width of retention time window 0.1 min; width of *m/z* window 30 ppm, split buckets with multiple compounds: yes, bucket filter: value count of bucket ≥ 10%, allow empty group attributes: yes, bucket value transformation: none and display bucket values in table: yes. The resulting bucket table consisted of 1420 buckets and served as a data matrix for the calculation of the following statistical models.

### 3.5. PLS Modeling

The bucket table (1420 bucket variables × 20 analyses) was imported to the software SIMCA 16.0.1 (Sartorius Stedim Data Analytics AB, Umeå, Sweden) and constituted the X-matrix (independent variables) of the PLS model. The bioactivity data of all fractions were transformed to logarithmic scale (pIC_50_ = −log IC_50_ [µM]) and used as the Y-matrix (dependent variables). Three different PLS models were generated with the activity data against *Pf* and *Tbr*, respectively, as well as the cytotoxicity values against mammalian cells (L6 cell line). All variables were scaled to unit variance (scaling factor: standard deviation) in order to assign equal weights to every variable in the model. The leave-one-out cross-validation was used to fully cross-validate the resulting models.

Buckets with a value ≥ 0.09 for w*c(1) (combined loading weight vector of X and Y for the first PLS component) and a w*c(2) ≤ 0.41 (combined loading weight vector of X and Y for the second PLS component) (pIC_50_ *Pf* 0.145381 w*c(1) and 0.237369 w*c(2)) were analyzed in the loadings plot of the PLS model with prediction of antiplasmodial compounds (Figure 1 and Figure 2). In the antitrypanosomal PLS model buckets with a w*c(1) ≥ 0.08 and a w*c(2) ≤ 0.13 (pIC_50_ *Tbr* 0.121195 w*c(1) and 0.119258 w*c(2)) were evaluated (Figure 4 and Figure 5).

### 3.6. In Vitro Bioassays

In vitro assays for the bioactivity of the 20 CPC fractions against *Tbr* (bloodstream trypomastigotes, STIB 900 strain) and *Pf* (intraerythrocytic forms, NF54 strain), and cytotoxicity tests against mammalian cells (L6-cell line from rat-skeletal myoblasts), were performed according to established standard protocols [18].

## 4. Conclusions

In conclusion, PLS models with good statistical performance were obtained with the LC/MS-data of 20 alkaloid fractions from the leaf extract of *B*. *sempervirens* L. for their antiplasmodial and antitrypanosomal activity as well as their cytotoxicity against mammalian cells. Using these models, antiprotozoal activity was predicted for eleven constituents. Of the four compounds responsible for the antiplasmodial activity, compound **1** and **4** have already been tested for their growth inhibition against *Pf* in the in vitro assay and exhibited promising IC_50_ values (IC_50_: 1.05 µM (**1**); IC_50_: 1.76 µM (**4**)). Compound **2** represents a previously undescribed natural product. This new *Buxus*-alkaloid is of interest for subsequent isolation and testing, especially because it is not predicted by the respective PLS model to significantly contribute to cytotoxicity.

Among the eight compounds highlighted by the PLS model for antitrypanosomal activity, compound **4** and **11** have already been tested against *Tbr* and displayed reasonable activity (IC_50_: 2.3 µM (**4**); IC_50_: 1.5 µM (**11**)). Moreover, two new natural products, compound **6** and **10**, were predicted to be effective against *Tbr*. Compound **8** could be dereplicated as *N*-formylcyclovirobuxeine-B which, to the best of our knowledge, is described for the first time as a constituent of *B*. *sempervirens* L. Since they did not significantly contribute to the PLS model for cytotoxicity, compounds **6** and **8** are the most interesting compounds for further studies in terms of antitrypanosomal *Buxus*-alkaloids.

The present study thus represents a new, impressive example showing that the chosen PLS approach is an auspicious means of identifying active natural products in a time- and resource-saving manner that can rationally guide the subsequent isolation of such compounds. Since the concentrations of compounds **2**, **6**, **8** and **10** in their respective fractions were quite low, their isolation will have to start from a larger amount of plant material which can now be performed in a target-oriented manner.

## Figures and Tables

**Figure 1 molecules-26-06181-f001:**
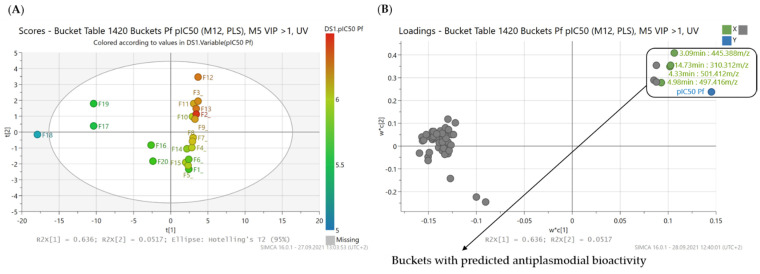
PLS model with prediction of antiplasmodial compounds from *B*. *sempervirens* L. (**A**) Scores plot of the second vs. the first PLS component of the PLS model. Coloration indicates the bioactivity against *Pf* of the fractions ranging from a pIC_50_ value of 5.15 (F18, blue) to a value of 6.46 (F2, red); (**B**) corresponding loadings plot to (**A**). The buckets marked with a box in the upper right quadrant are predicted as compounds with high antiplasmodial activity by the PLS model (Buckets marked and labelled in green are detailed in Figure 2).

**Figure 2 molecules-26-06181-f002:**
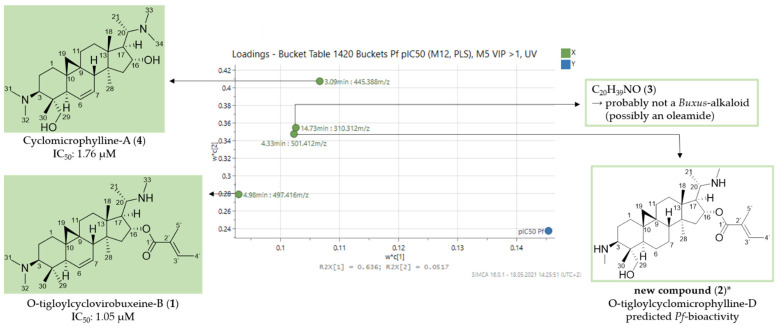
Expansion of region of the loadings plot (Figure 1) with the buckets that, according to the PLS model, have a high impact on the antiplasmodial activity of *B*. *sempervirens* L. * Structural assignment based only on mass spectral data; stereochemistry tentatively assumed in analogy with known compounds [7].

**Figure 3 molecules-26-06181-f003:**
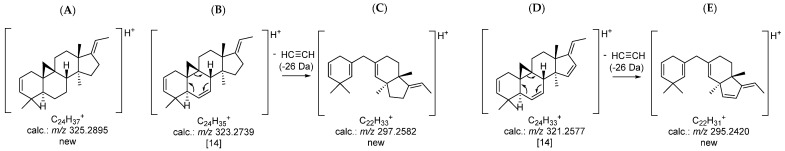
Prominent core fragments of the investigated *nor*-triterpene alkaloids of the *nor*-cycloartane type from *B*. *sempervirens* L. (**A**) core fragment *m/z* 325 for fully saturated C-3/C-20 diamines; (**B**) core fragment *m/z* 323 for C-3/C-20 diamines possessing one additional substitution or double bond; (**C**) core fragment *m/z* 297 for C-3/C-20 diamines with a Δ^6^; (**D**) core fragment *m/z* 321 for C-3/C-20 diamines doubly substituted or unsaturated; (**E**) core fragment *m/z* 295 for C-3/C-20 diamines with a Δ^6^ and one further substitution or unsaturation. (The postulated positions of unsaturation are based on the already known structures of compounds **1** and **4**).

**Figure 4 molecules-26-06181-f004:**
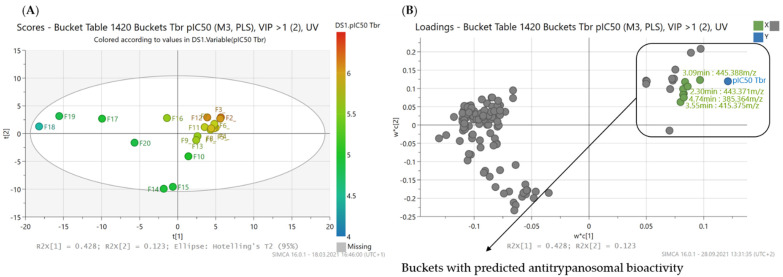
PLS model with prediction of antitrypanosomal compounds from *B*. *sempervirens* L. (**A**) Scores plot of the second vs. the first PLS component of the PLS model. Coloration indicates the bioactivity against *Tbr* of the fractions ranging from a pIC_50_ value of 4.41 (F18, blue) to a value of 6.2 (F2, red); (**B**) corresponding loadings plot to (**A**). The buckets marked with a box in the upper right quadrant are predicted to possess high antitrypanosomal activity by the PLS model (Buckets marked and labeled in green are detailed in Figure 5).

**Figure 5 molecules-26-06181-f005:**
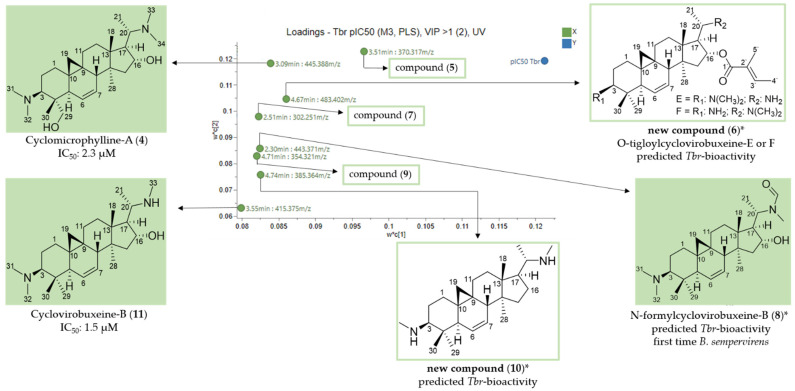
Expansion of the region of the loadings plot (Figure 4) with the buckets that, according to the PLS model, have a high impact on the antitrypanosomal activity of *B*. *sempervirens* L. * Assignment based only on mass spectral data; stereo-chemistry tentatively assumed in analogy with known compounds [7].

**Figure 6 molecules-26-06181-f006:**
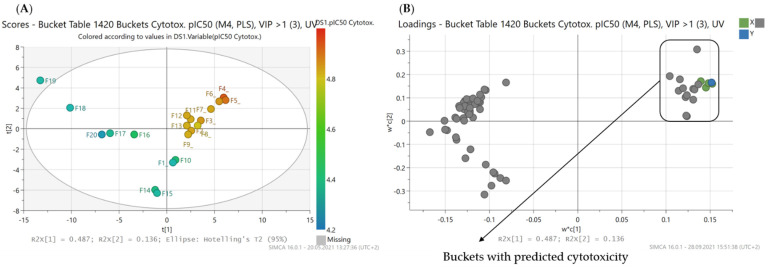
PLS model with a prediction of cytotoxic compounds from *B*. *sempervirens* L. (**A**) Scores plot of the second vs. the first PLS component of the PLS model. Coloration indicates the cytotoxicity of the fractions ranging from a pIC_50_ value of 4.26 (F20, blue) to a value of 4.94 (F4, red); (**B**) corresponding loadings plot to (**A**). The buckets marked with a box in the upper right quadrant are predicted as compounds mainly responsible for the low overall cytotoxic activity of the *B. sempervirens* extract (Buckets marked in green are detailed in Figure 7).

**Figure 7 molecules-26-06181-f007:**
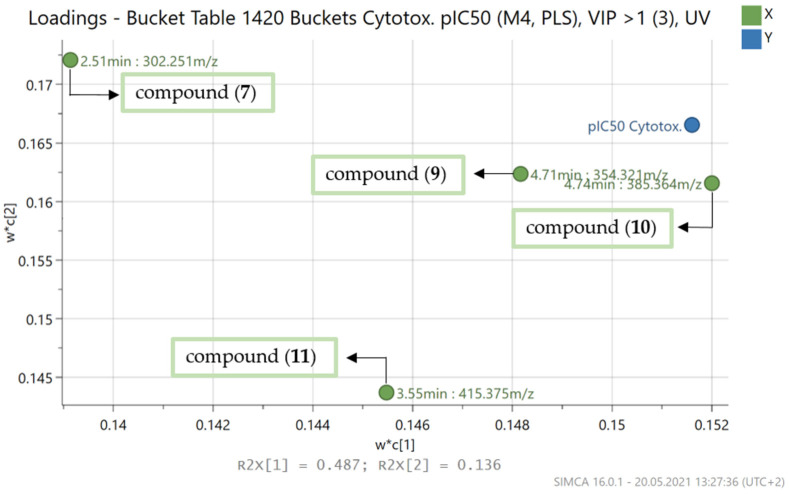
Expansion of region of the loadings plot (Figure 6) with the buckets that, according to the PLS model, have a high impact on the cytotoxicity of *B*. *sempervirens* L.

**Table 1 molecules-26-06181-t001:** In vitro antiprotozoal and cytotoxic activity of the crude dichloromethane (DCM) extract and fractions from *B*. *sempervirens* L. leaves.

Sample	*Pf*	*Tbr*	Cytotox.	SI *Pf*	SI *Tbr*
GBUS	7.9 ± 0.06	14.4 ± 2.7	56.4 ± 1.7	7	4
LNB	16 ± 6.1	15 ± 5	54 ± 4	3	4
ALOF	0.99 ± 0.03	5 ± 1.4	34 ± 11.7	34	7
F1	1.5 ± 0.2	1.98 ± 0.5	48	32	24
F2	0.35 ± 0.01	0.63 ± 0.03	15	43	24
F3	0.55 ± 0.04	0.92 ± 0.06	14	26	15
F4	0.92 ± 0.03	1.4 ± 0.3	12	13	9
F5	0.99 ± 0.004	1.4 ± 0.04	13	13	9
F6	1.5 ± 0.4	1.7 ± 0.7	14	9	8
F7	0.75 ± 0.04	1.8 ± 0.6	15	20	8
F8	0.76 ± 0.07	1 ± 0.3	16	21	16
F9	0.62 ± 0.35 *	2.4 ± 1.41	15	24	6
F10	0.95 ± 0.19	9.8 ± 5.3	39	41	4
F11	0.65 ± 0.25	1.8 ± 0.72	15	23	8
F12	0.5 ± 0.23 *	0.8 ± 0.04	15	30	19
F13	0.49 ± 0.21 *	2.6 ± 0.9	15	31	6
F14	1.18 ± 0.54 *	12 ± 1	41	35	3
F15	1.0 ± 0.26	12 ± 0.1	46	46	4
F16	1.82 ± 0.53	2.7 ± 0.2	37	20	14
F17	3.0 ± 0.5	11 ± 1	44	15	4
F18	7.11 ± 1.8	39.3 ± 0.7	48	7	1
F19	2.67 ± 0.2	14.4 ± 1.7	47	18	3
F20	1.9 ± 0.7	13.5 ± 1	55	29	4
Chloroquine	0.003 ± 0.001			3	
Melarsoprol		0.004 ± 0.001			3
Podophyllotoxin			0.01 ± 0.004		

GBUS: crude dichloromethane extract of *B*. *sempervirens* L. leaves; LNB: lipophilic and neutral fraction; ALOF: alkaloid fraction; SI: selectivity index. All IC_50_ values are expressed in µg/mL. While * *n* = 3 was reported as the mean value from three independent measurements with the standard deviation, the other values were determined with *n* = 2 as the mean value from two independent measurements with the fluctuation range and *n* = 1 for the cytotoxicity data.

**Table 2 molecules-26-06181-t002:** Statistical characteristics of the three different PLS models.

PLS Model	PLS Components	R^2^	Q^2^ *
*Pf*	2	0.9	0.82
*Tbr*	4	0.98	0.89
Cytotox.	3	0.96	0.83

* leave-one-out cross validation.

**Table 3 molecules-26-06181-t003:** LC/MS-characteristics of the compounds with predicted antiplasmodial activity (compare Figure 2).

Bucket	Adduct Ions	Structural Formula (DBE)	Core Fragment(s) *m/z*	Identified *Buxus*-Alkaloid
4.98 min: 497.416 *m/z* (**1**)	[M + H]^+^ < [M + 2H]^2+^	C_32_H_52_N_2_O_2_ (8)	321; 295	*O*-tigloylcyclovirobuxeine-B
4.33 min: 501.412 *m/z* (**2**)	[M + H]^+^ < [M + 2H]^2+^	C_31_H_52_N_2_O_3_ (7)	323	new compound
14.73 min: 310.312 *m/z* (**3**)	[M + H]^+^	C_20_H_39_NO (2)	n.i.	n.i.; possibly an oleamide structure
3.09 min: 445.388 *m/z* (**4**)	[M + H]^+^ < [M + 2H]^2+^	C_28_H_48_N_2_O_2_ (6)	321; 295	cyclomicrophylline-A

DBE: double bond equivalent; n.i.: not identified.

**Table 4 molecules-26-06181-t004:** LC/MS-characteristics of the compounds with predicted antitrypanosomal activity (compare Figure 5).

Bucket	Adduct Ions	Structural Formula (DBE)	Core Fragment(s) *m/z*	Identified *Buxus*-Alkaloid
3.51 min: 370.317 *m/z* (**5**)	[M + H]^+^	C_25_H_39_NO (7)	n.i.	n.i.
3.09 min: 445.388 *m/z* (**4**)	[M + H]^+^ < [M + 2H]^2+^	C_28_H_48_N_2_O_2_ (6)	321; 295	cyclomicrophylline-A
4.67 min: 483.402 *m/z* (**6**)	[M + H]^+^ < [M + 2H]^2+^	C_31_H_50_N_2_O_2_ (8)	321; 295	new compound
2.51 min: 302.251 *m/z* (**7**)	[M + H]^+^ < [M + 2H]^2+^	C_20_H_31_NO (6)	n.i.	n.i.
2.30 min: 443.371 *m/z* (**8**)	[M + H]^+^ < [M + 2H]^2+^	C_28_H_46_N_2_O_2_ (7)	321; 295	*N*-formylcyclovirobuxeine-B
4.71 min: 354.321 *m/z* (**9**)	[M + H]^+^ < [M + 2H]^2+^	C_25_H_39_N (7)	n.i.	n.i.
4.74 min: 385.364 *m/z* (**10**)	[M + H]^+^ < [M + 2H]^2+^	C_26_H_44_N_2_ (6)	323; 297	new compound
3.55 min: 415.375 *m/z* (**11**)	[M + H]^+^ < [M + 2H]^2+^	C_27_H_46_N_2_O (6)	321; 295	cyclovirobuxeine-B

DBE: double bond equivalent; n.i.: not identified.

## Data Availability

Data is contained within the article and Appendix A. The underlying raw data are available from the corresponding author on request.

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
