# Peer review of "Identification of Antiprotozoal Compounds from *Buxus* *sempervirens* L. by PLS-Prediction"

_molecules, 2021, doi:10.3390/molecules26206181_

Round 1
Reviewer 1 Report
You have some interesting science and ideas.
As a minor point, you have an overuse of abbreviations. I don't understand why you have all of these made up abbreviations. It makes it hard to read your paper. It interrupts the flow. It makes the content very hard to skim, as you have to remember what your non-standard abbreviations are. I suggest that you re-think the need for abbreviations like: ALOF, GBUS, Cytotox., etc.
As a major point, I fell like the story is not complete. At the end of the paper, you cannot really say what the compounds are. You can point to figures and dots. You can show some LC-MS data. I don't understand why you simply did not isolate the compounds and solve their structures. It's like the story is 80% complete, with 20% more to do to isolate the molecules and then run/interpret the NMR data. This is likely a point where we disagree, since you decided to stop the paper where you did. I simply don't understand. The conclusions seem unsatisfying to me.
Other things to consider:
With finding already known active metabolites through PLS modelling, the authors showed that the methodology can be successfully applied. However, when identifying novel metabolites, it should be carefully applied to structure elucidation. MS and MS/MS data alone, should not be used to describe the structure of a novel metabolite, instead it should be a tentative or suggested structure since these compounds were not isolated and examined through traditional structure elucidation methods (e.g. through NMR spectroscopy). The paper shows that there is interest in examining Buxus sempervirens L. closely. This paper would be much stronger if the suggested novel metabolites would be isolated. Even though there is not actual isolation work in this paper, I believe this could be published in this journal with some changes and fixes listed below.
Page 2. Line 54 – Is that sentence refers to this paper or a previous study? Because there were only 20 fractions not 25 alkaloids and there weren’t any isolated just semi purified. Please be precise when you say „in this study”
Page 3. Line 87- Are these abbreviations necessary? It just confusing me and does not abbreviate super long words. I think it can be kept as alkaloid rich fraction and crude extract instead of (GBUS and ALOF).
Page 3. Line 95- I would rather say implied than clearly showed. Since none of the compounds were isolated.
Page 3. Line 118- All of the supplementary MS base peak chromatograms look ugly by having the PDA chromatogram and MS data mashed into one graph. Can it be separated? I have never seen a paper representing it like this.
Figure 1.- Is there a way to only show the retention time m/z values on the loading plot for those that are being examined in this paper? Showing all the features makes the loading plot a little bit too crowded and hard to read since labels are overlapping. Is the loading figure cropped from another image? Is it a high-resolution image? I can see above the sentence “bucket with predicted antiplasmodial bioactivity” that there are some residues from the original image it was cropped.
Page 6. Line 194- I would refer to this structure as tentatively identified or tentative structure. The structure was solved based upon MS and MS/MS data. Line 196- Authors note that the stereochemistry was assumed based upon the known analogue. Would it be better to not to show the stereochemistry at all on Figure 2 then? There is no scientific evidence for such assumption. I wouldn’t give a new scientific name to metabolite that was only identified based on MS data.
Figure 4. Loading plot is hard to read again, because of the overlapping labels of the features. Could it be cleaned up?
Figure 5. On previous figures such as Figure 2. The nomenclature and numbering were consistent. Each dot on the loading plot got an increasing number starting from the bottom to the top, referring to 4 compounds. On this figure the author jumps between dots and numbering inconsistently. It is hard to follow which dot is which compound based on the numbering. It also makes it hard to compare this figure with Table 4. I understand that compound 4 is on both loading plot but the author should keep the numbering consistent on Figure 5 too.
Author Response
Response to Reviewer 1:
You have some interesting science and ideas.
As a minor point, you have an overuse of abbreviations. I don't understand why you have all of these made up abbreviations. It makes it hard to read your paper. It interrupts the flow. It makes the content very hard to skim, as you have to remember what your non-standard abbreviations are. I suggest that you re-think the need for abbreviations like: ALOF, GBUS, Cytotox., etc.
Reply:
We have tried to minimize the number of abbreviations. However, since all of them are properly explained in the first instance of mentioning, we believe it is ok to keep abbreviations instead of always spelling out the full terms.
As a major point, I fell like the story is not complete. At the end of the paper, you cannot really say what the compounds are. You can point to figures and dots. You can show some LC-MS data. I don't understand why you simply did not isolate the compounds and solve their structures. It's like the story is 80% complete, with 20% more to do to isolate the molecules and then run/interpret the NMR data. This is likely a point where we disagree, since you decided to stop the paper where you did. I simply don't understand. The conclusions seem unsatisfying to me.
Reply:
We are sorry if the reviewer misunderstood this. The story is complete in that the main alkaloids were already isolated and published in our previous paper (now reference [7]). However, the alkaloids with high predicted activity by the PLS models of the present study would be very interesting to isolate and test. However, since these compounds are present only in small amounts in the respective fractions, a target-guided isolation of these alkaloids with a new and larger batch of boxwood leaves would be necessary to obtain them in sufficient quantities. This is, unfortunately, not possible at present but will certainly be done in the future. A short statement clarifying this has been added at the end of the conclusions.
Other things to consider:
With finding already known active metabolites through PLS modelling, the authors showed that the methodology can be successfully applied. However, when identifying novel metabolites, it should be carefully applied to structure elucidation. MS and MS/MS data alone, should not be used to describe the structure of a novel metabolite, instead it should be a tentative or suggested structure since these compounds were not isolated and examined through traditional structure elucidation methods (e.g. through NMR spectroscopy). The paper shows that there is interest in examining Buxus sempervirens L. closely. This paper would be much stronger if the suggested novel metabolites would be isolated. Even though there is not actual isolation work in this paper, I believe this could be published in this journal with some changes and fixes listed below.
Reply:
We have clearly marked our structural assignment as tentative (see legends of figures 2 and 5 and hope this will be ok with the reviewer. The assignments appear the most plausible ones based on the structures of a large number of known alkaloids of this type from Buxus.
Page 2. Line 54 – Is that sentence refers to this paper or a previous study? Because there were only 20 fractions not 25 alkaloids and there weren’t any isolated just semi purified. Please be precise when you say „in this study”.
Reply:
This has been corrected to a less ambiguous statement.
Page 3. Line 87- Are these abbreviations necessary? It just confusing me and does not abbreviate super long words. I think it can be kept as alkaloid rich fraction and crude extract instead of (GBUS and ALOF).
Reply:
This has been corrected in the continuous text, partly. As the abbreviations are consistent with our two previous publications and are all defined in the manuscript, we would like to retain them, for instance in Table 1.
Page 3. Line 95- I would rather say implied than clearly showed. Since none of the compounds were isolated.
Reply:
This has been corrected.
Page 3. Line 118- All of the supplementary MS base peak chromatograms look ugly by having the PDA chromatogram and MS data mashed into one graph. Can it be separated? I have never seen a paper representing it like this.
Reply:
The chromatograms have been separated.
Figure 1.- Is there a way to only show the retention time m/z values on the loading plot for those that are being examined in this paper? Showing all the features makes the loading plot a little bit too crowded and hard to read since labels are overlapping. Is the loading figure cropped from another image? Is it a high-resolution image? I can see above the sentence “bucket with predicted antiplasmodial bioactivity” that there are some residues from the original image it was cropped.
Reply:
Figure 1 has been cleaned up and has been provided in higher resolution.
Page 6. Line 194- I would refer to this structure as tentatively identified or tentative structure. The structure was solved based upon MS and MS/MS data. Line 196- Authors note that the stereochemistry was assumed based upon the known analogue. Would it be better to not to show the stereochemistry at all on Figure 2 then? There is no scientific evidence for such assumption. I wouldn’t give a new scientific name to metabolite that was only identified based on MS data.
Reply:
To clarify that the structure elucidation was carried out using MS and MS/MS data, footnotes have been added to figures 2 and 5, respectively, also with a respect to the tentative assignment of stereochemistry. The stereochemical assumptions appear the most plausible ones based on the structures of a large number of known alkaloids of this type from Buxus.
Since there is an existing classification for naming the Buxus-alkaloids [15, 16], we just quote what the postulated structures would be systematically called. These are not newly invented trivial names.
Figure 4. Loading plot is hard to read again, because of the overlapping labels of the features. Could it be cleaned up?
Reply:
Figure 4 has been cleaned up.
Figure 5. On previous figures such as Figure 2. The nomenclature and numbering were consistent. Each dot on the loading plot got an increasing number starting from the bottom to the top, referring to 4 compounds. On this figure the author jumps between dots and numbering inconsistently. It is hard to follow which dot is which compound based on the numbering. It also makes it hard to compare this figure with Table 4. I understand that compound 4 is on both loading plot but the author should keep the numbering consistent on Figure 5 too.
Reply:
In order to make it uniform and to facilitate the allocation to Table 4, all dots were assigned to Figure 5. In our view, the numbering is consistent, since in both figures 2 and 5 the numbering started with the compound closest to the Y variable and then continued in descending order.
We thank the reviewer for the time and effort to help us improve our manuscript.
Reviewer 2 Report
The work by Szabo et al. describes a unique way of analyzing complex data from multiple sources to gain meaningful biological insight. In their case, they are using LC/MS data acquired on natural products from B. sempervirens, bioactivity and cytotoxicity to identify compounds with antiprotozoal activity. The paper is nicely written and showcase a statistical approach to find novel compounds of interest. The manuscript is deemed fit for publication following minor edits.
Specific Comments:
- Lines 40-51 : This paragraph from the authors is very important to highlight the need for identification of novel anti-protozoal agents. Add appropriate references to strengthen your statements.
- Figure1, 4 and 6: While the top right quadrant in the scores plot and the corresponding loadings plot show lot more hits compared to the ones discussed by the authors. Does it mean, there were lot more m/z values from the bucket table that could not be annotated. Please add a brief statement either in the main text or Materials and Method section describing your filtering criteria. It will also be helpful, if authors can provide a clearer version of the loading plot in all these figures.
- Line 180: What do authors mean by authentic samples?
- Mass tolerance: It is not clear what ppm mass error threshold was used for annotating peaks with their respective compounds.
- While authors, mention about sharing RAW data on request, it will be very helpful if they can upload the annotated input tables (bucket tables and bioactivity data) used for PLS analysis, so that broad scientific community can benefit from this rich dataset.
Author Response
Response to Reviewer 2
The work by Szabo et al. describes a unique way of analyzing complex data from multiple sources to gain meaningful biological insight. In their case, they are using LC/MS data acquired on natural products from B. sempervirens, bioactivity and cytotoxicity to identify compounds with antiprotozoal activity. The paper is nicely written and showcase a statistical approach to find novel compounds of interest. The manuscript is deemed fit for publication following minor edits.
Specific Comments:
- Lines 40-51 : This paragraph from the authors is very important to highlight the need for identification of novel anti-protozoal agents. Add appropriate references to strengthen your statements.
Reply:
Appropriate references have been added.
- Figure1, 4 and 6: While the top right quadrant in the scores plot and the corresponding loadings plot show lot more hits compared to the ones discussed by the authors. Does it mean, there were lot more m/z values from the bucket table that could not be annotated. Please add a brief statement either in the main text or Materials and Method section describing your filtering criteria. It will also be helpful, if authors can provide a clearer version of the loading plot in all these figures.
Reply:
The filtering criteria have been described (3.5 Materials and Methods section).
Figures have been cleaned up and have been provided in higher resolution.
- Line 180: What do authors mean by authentic samples?
Reply:
This has been clarified.
- Mass tolerance: It is not clear what ppm mass error threshold was used for annotating peaks with their respective compounds.
Reply:
This information has been added (3.3 Materials and Methods section).
- While authors, mention about sharing RAW data on request, it will be very helpful if they can upload the annotated input tables (bucket tables and bioactivity data) used for PLS analysis, so that broad scientific community can benefit from this rich dataset.
Reply:
The data tables with X and Y variables of the three PLS models are provided in csv format.
We thank the reviewer for the time and effort to help us improve our manuscript.

Reviewer 3 Report
The author used UHPLC/+ESI-QqTOF-MS/MS to analyze 20 compounds extracted from Buxus (B.) sempervirens L., and based on PLS analysis, their spectral signals were compared with their cytotoxicity and the inhibitory activity of the two plasmodium , established a screening system for screening compounds with strong activity and low cytotoxicity, successfully used it to screen out multiple compounds, and verified their activity through literature research. It includes three unknown ingredients, which may be new natural products.
This article is rigorous in the construction of the PLS model, and the analysis and discussion of the correlation between multiple variables are detailed. I only have one request to make about this journal.
It is suggested that the author can supplement to design a comprehensive screening model based on PLS analysis, which can simultaneously model the spectrum data of each compound and the inhibitory activity of the two malaria parasites and their cytotoxicity, so as to be able to comprehensively screen for the compounds who has good activity against both malaria parasites at the same time and less cytotoxic , rather than modeling separately.
Author Response
Response to Reviewer 3
The author used UHPLC/+ESI-QqTOF-MS/MS to analyze 20 compounds extracted from Buxus (B.) sempervirens L., and based on PLS analysis, their spectral signals were compared with their cytotoxicity and the inhibitory activity of the two plasmodium , established a screening system for screening compounds with strong activity and low cytotoxicity, successfully used it to screen out multiple compounds, and verified their activity through literature research. It includes three unknown ingredients, which may be new natural products.
This article is rigorous in the construction of the PLS model, and the analysis and discussion of the correlation between multiple variables are detailed. I only have one request to make about this journal.
It is suggested that the author can supplement to design a comprehensive screening model based on PLS analysis, which can simultaneously model the spectrum data of each compound and the inhibitory activity of the two malaria parasites and their cytotoxicity, so as to be able to comprehensively screen for the compounds who has good activity against both malaria parasites at the same time and less cytotoxic , rather than modeling separately.
Reply:
We thank the reviewer for the positive assessment and the valuable suggestion for further studies using, possibly, other statistical methods which were not within the scope of our present study.

Round 2
Reviewer 1 Report
Your abbreviations being consistent between previous papers that you published is complete rubbish. No one uses those abbreviations for alkaloid rich fractions, and alkaloids have been studied since the 1800s. You are only harming yourself by sticking to those.
Reviewer 3 Report
The authors have revised the manuscript according to the suggestions of the reviewers, I think it suitable to be accepted.